# Treatment with a Cholecystokinin Receptor Antagonist, Proglumide, Improves Efficacy of Immune Checkpoint Antibodies in Hepatocellular Carcinoma

**DOI:** 10.3390/ijms24043625

**Published:** 2023-02-11

**Authors:** Narayan Shivapurkar, Martha D. Gay, Aiwu (Ruth) He, Wenqiang Chen, Shermineh Golnazar, Hong Cao, Tetyana Duka, Bhaskar Kallakury, Sona Vasudevan, Jill P. Smith

**Affiliations:** 1Department of Medicine, Georgetown University, Washington, DC 20007, USA; 2Department of Oncology, Georgetown University Lombardi Comprehensive Cancer Center, Washington, DC 20007, USA; 3Department of Biochemistry and Molecular & Cellular Biology, Georgetown University, Washington, DC 20057, USA; 4Department of Pathology, MedStar Georgetown University Hospital, Washington, DC 20007, USA

**Keywords:** hepatocellular carcinoma, cholecystokinin receptor, proglumide, tumor microenvironment, immune checkpoint antibodies, epithelial-to-mesenchymal transition

## Abstract

Hepatocellular carcinoma (HCC) is the third leading cause of cancer-associated deaths worldwide. Treatment with immune checkpoint antibodies has shown promise in advanced HCC, but the response is only 15–20%. We discovered a potential target for the treatment of HCC, the cholecystokinin-B receptor (CCK-BR). This receptor is overexpressed in murine and human HCC and not in normal liver tissue. Mice bearing syngeneic RIL-175 HCC tumors were treated with phosphate buffer saline (PBS; control), proglumide (a CCK-receptor antagonist), an antibody to programmed cell death protein 1 (PD-1Ab), or the combination of proglumide and the PD-1Ab. In vitro, RNA was extracted from untreated or proglumide-treated murine Dt81Hepa1-6 HCC cells and analyzed for expression of fibrosis-associated genes. RNA was also extracted from human HepG2 HCC cells or HepG2 cells treated with proglumide and subjected to RNA sequencing. Results showed that proglumide decreased fibrosis in the tumor microenvironment and increased the number of intratumoral CD8+ T cells in RIL-175 tumors. When proglumide was given in combination with the PD-1Ab, there was a further significant increase in intratumoral CD8+ T cells, improved survival, and alterations in genes regulating tumoral fibrosis and epithelial-to-mesenchymal transition. RNAseq results from human HepG2 HCC cells treated with proglumide showed significant changes in differentially expressed genes involved in tumorigenesis, fibrosis, and the tumor microenvironment. The use of the CCK receptor antagonist may improve efficacy of immune checkpoint antibodies and survival in those with advanced HCC.

## 1. Introduction

For more than 10 years, sorafenib was the only systemic therapy approved by the Food and drug administration (FDA) for the treatment of hepatocellular carcinoma (HCC) [1]. Tyrosine kinase inhibitors, such as sorafenib, lenvatinib, regorafenib, and cabozantinib, may extend survival for a few months but are not curative [2]. In recent years, immune checkpoint inhibitor (ICI)-based regimens have become standard of care in the first-line setting, as well as for the treatment of disease that has progressed on prior sorafenib treatment [3]. The CheckMate 040 and 459 trials demonstrated a survival benefit of nivolumab in treatment-naive or previously treated patients with sorafenib, although it was not superior to the first-line therapy, sorafenib [4,5]. A second PD-1Ab, pembrolizumab, was then approved as a second-line therapy in 2018 after the Keynote 224 trial showed positive results in patients with previously treated advanced HCC [6]. In 2020, the CheckMate 040 study showed that nivolumab in combination with ipilimumab, a cytotoxic T-lymphocyte-associated protein (CTLA-4) immune checkpoint inhibitor, had an overall response rate of 33% in previously treated patients with advanced HCC and was approved as a second-line therapy [7]. Shortly thereafter, the combination therapy, atezolizumab, a PD-L1 inhibitor, with bevacizumab, a vascular endothelial growth factor inhibitor, demonstrated a significant survival benefit with a medium overall survival (OS) of 19 months over the standard therapy, sorafenib, in the IMBrave 150 trial [8]. This combination then received approval as a first-line therapy for unresectable HCC based on the results of this phase 3 trial [9]. The second ICI regimen to receive full approval for the treatment of HCC was tremelimumab in combination with durvalumab for patients with unresectable HCC, which was approved in 2022. Approval was based on the open-label, phase III HIMALYA trial, in which 1171 patients were randomly assigned (1:1:1) to receive tremelimumab in combination with durvalumab (*n* = 393), durvalumab (*n* = 389), or sorafenib (*n* = 389). The primary endpoint for this trial was OS. Median OS for the tremelimumab in combination with durvalumab arm was 16.43 months (95% confidence interval (CI) 14.16 to 19.58), compared to 16.56 months (95% CI 14.06 to 19.12) in the durvalumab arm and 13.77 months (95% CI 12.25 to 16.13) in the sorafenib arm [9]. As of 2022, two ICI-based combinations, atezolizumab with bevacizumab [8] and tremelimumab with durvalumab [10], had full FDA approval for first-line treatment of HCC, and pembrolizumab monotherapy [6] as well as nivolumab in combination with ipilimumab [7] had accelerated approvals as second-line options.

Despite the progress made in the systemic therapy of advanced HCC, only a minority of patients experience tumor shrinkage to current immunotherapies and challenges have been met with finding the best combination regimen [11]. In addition, there is no biomarker that can identify the group of subjects that would be responders to immunotherapy [12]. Although there are nine FDA-approved treatments, no strategy is developed to sequentially order the treatment for best clinic outcome. Early deaths have been reported with ICI monotherapy compared to administration of ICIs with other agents [13]. For these reasons, new strategies for HCC therapy are needed that are target-specific with limited toxicity.

Our research laboratory discovered that the cholecystokinin-B receptor (CCK-BR) is absent in normal liver tissue but becomes overexpressed in HCC, suggesting that this receptor may be an ideal target for the treatment of HCC [14]. Human HepG2 HCC cells express high levels of CCK-BR mRNA. When HepG2 cells were stimulated with a ligand for the CCK-BR (gastrin), the cells increased in proliferation by the clonogenic assay and this effect was blocked by the CCK-BR antagonist proglumide [14]. Murine HCC cell lines (RIL-175 [15] and Dt81Hepa1-6 [16]) also express CCK-BRs [14]. Furthermore, monotherapy with proglumide was shown to significantly decrease the growth of HCC tumors in mice [14].

In the current investigation, we studied the effects of the CCK-BR antagonist, proglumide in combination with an immune checkpoint antibody, PD-1Ab, on the growth of HCC tumors in mice. Furthermore, we explored the effects of proglumide therapy on differentially expressed genes in murine RIL-175 HCC tumors and murine Dt81Hepa1-6 and human HepG2 HCC cells.

## 2. Results

### 2.1. Receptor and Gastrin Expression in HCC Cells

RNA was extracted from murine RIL-175 HCC cells and subjected to qRT-PCR for gene expression of CCK-ARs, CCK-BRs, gastrin peptide, and PD-L1 receptor expression. Compared to benign normal mouse liver, RIL-175 murine HCC cells exhibited increased expression of the CCK-AR and markedly increased expression of the CCK-BR (Figure 1A). The endogenous gastrin peptide mRNA was markedly increased in murine HCC cells (Figure 1B).

Gastrin is normally made in the G cells of the stomach antrum [17], but aberrant synthesis of gastrin is found in some gastrointestinal cancers, including pancreatic cancer, where gastrin stimulates growth by an autocrine mechanism [18]. The PD-L1 receptor expression was also increased in RIL-175 HCC cancer cells compared to normal mouse liver by qRT-PCR (Figure 1B). PCR dissociation curves for mouse CCK-BR and gastrin are shown in Appendix A.

### 2.2. HCC Tumor Growth and Mouse Survival

We studied the effects of a PD-1Ab alone, proglumide alone, or the combination of a PD-1Ab with proglumide on growth of RIL-175 murine syngeneic tumors in C57BL/6 mice. Proglumide monotherapy was as effective as the PD-1Ab in inhibiting growth of HCC and, when proglumide and the PD-1Ab were administered in combination, tumor volumes were significantly lower than tumors of control mice (Figure 2A). Only 50% of the control mice were alive at 4 weeks (Figure 2B) but 100% of the mice treated with the combination of proglumide and the PD-1Ab were still alive, suggesting a significant survival benefit with the combination therapy.

### 2.3. Proglumide Alters the Tumor Microenvironment by Decreasing Fibrosis and Increasing CD8+ T Cells

Immunohistochemistry was performed on tumor sections from each treatment group or control tumors for CD8+ T lymphocytes. Figure 3A shows representative photos of RIL-175 tumors from each treatment group. There was a paucity of CD8+ T cells in tumors of control mice and in the tumors of the PD-1Ab-treated mice (Figure 3B). In contrast, the number of CD8+ cells was visibly greater in the tumors of mice treated with proglumide (Figure 3B). The number of immunoreactive CD8+ cells was significantly increased in the tumors of mice receiving proglumide therapy, and there was a significant synergistic effect on the number of CD8+ intratumoral T cells when the two agents (proglumide and the PD-1Ab1 ab) were combined (*p* = 1.3 × 10^−8^).

RIL-175 mouse tumors were stained with Masson’s trichrome to study the degree of fibrosis in the tumor microenvironment. A representative image from each treatment group reveals tumor fibrosis (Figure 3C). The integrative density analysis of tumor fibrosis from the different treated groups showed that the PD-1Ab monotherapy did not alter tumor fibrosis. Proglumide monotherapy and the combination-therapy-treated mice exhibited a marked decrease in tumoral fibrosis (Figure 3D) compared to controls, and also compared to PD-1Ab monotherapy.

### 2.4. Differentially Expressed Genes in Tumors of Mice Treated with Proglumide and PD-1Ab

Total RNA was extracted from the control and combination-treated RIL-175 tumors and analyzed using a mouse liver cancer PCR array to identify differentially expressed genes (DEGs). A summary of top 10 genes from our qRT-PCR analysis and the PCR array that were significantly up- or downregulated in tumors of the mice treated with the combination of proglumide and PD-1Ab compared to control tumors is shown in Table 1, including a description of their function and a PubMed reference number. Many of the DEGs were involved in ferroptosis (*Ptgs2*), enhanced T-cell function (*Lef1*), regulators of EMT (*Zeb1, Zeb2, Vimentin,* and *Sfrp2*), and fibrosis (*Col1α1* and *Acta2*) in the tumor microenvironment.

In our previous study with a mouse model of pancreatic cancer [19], we observed a synergistic suppressive effect on tumor growth, fibrosis, and expression of genes implicated in epithelial-to-mesenchymal transition (EMT) and fibrosis when gemcitabine was combined with proglumide. Thus, we explored the effect of PD-1Ab, proglumide and the combination on those key genes identified that were associated specifically with tumoral EMT (Zinc Finger E-Box Binding Homeobox 1 and 2: *Zeb1 and Zeb2*), fibrosis (collagen type 1α chain 1: *Col1α1*), and (α-smooth muscle actin: *Acta2*). Using qRT-PCR, we found that expression of *Zeb1* and *Zeb2* was not significantly impacted by the treatment of the PD-1Ab or proglumide monotherapy (Figure 4A and Figure 4B, respectively); however, when these compounds were given together, there was a significant decrease in EMT-associated gene mRNA expression in the tumors, suggesting that the combination therapy decreases the potential to undergo EMT and metastasize. Expression of another gene involved in the EMT pathway, *Vimentin,* was also significantly decreased in the tumors of mice treated with proglumide alone or in combination with the PD-1Ab (Figure 4C). Since proglumide therapy decreased fibrosis of the tumors and an enhanced effect was seen with the combination therapy (above in Figure 3D), we further investigated the effects of the treatments on two genes (*Col1α1* and *Acta2)* involved with stellate cell or fibroblast activation. mRNA expression of these two fibrosis-associated genes showed significant downregulation in the tumors of mice treated with the combination therapy (Figure 4D and Figure 4E, respectively). Representative dissociation curve analysis of qPCR amplification is provided in the Appendix A. The raw cycle threshold (Ct) data detailed analysis of relative expression of these DEGs and statistical analysis is provided in the Appendix A. The genes involved in hepatic fibrosis identified in the liver cancer PCR array were further analyzed with Ingenuity Pathway Analysis (IPA) software (https://digitalinsights.qiagen.com/products-overview/discovery-insights-portfolio/analysis-and-visualization/qiagen-ipa/)and the key genes altered by treatment of mice with proglumide are highlighted in pink color in Appendix A.

### 2.5. Proglumide Therapy Significantly Decreases Fibrosis-Associated Genes in Dt81Hepa1-6 Cells In Vitro

We conducted a confirmatory study in vitro to analyze the effects of proglumide monotherapy (20 nM and 40 nM) on some of the above DEGs in another murine liver cancer cell line (Dt81Hepa1-6) after exposure for 8h and 24 h using qRT-PCR. Proglumide therapy also significantly downregulated *Col1α1,* even at a very early time point, (Figure 5) in this second HCC cell line. Similar to the RIL-175 tumors, no significant effect in the *Acta2* and EMT-associated genes *Zeb1* and *Zeb2* was seen with proglumide therapy alone in the Dt81Hepa1-6 HCC cells (Appendix A). As we also observed in the RIL-175 tumors, it appears that the greatest effects on the EMT genes occur when proglumide is given in combination with the PD-1Ab.

It is important to note that *Col1α1* is one of the most significantly downregulated genes following treatment of liver cancer cells in culture with proglumide and also significantly suppressed in liver tumors in mice following treatment with combination of an PD-1Ab and proglumide. It is possible genes like *Col1α1* are more proximal to the downstream effect of proglumide. Perhaps some of the other DEGs in our study found in the tumors may have a secondary effect or require the interaction with tumor microenvironment, which exists within tumors to exhibit a change with the treatment.

### 2.6. Characterization of the CCK-B Receptor in a Human Tissue Array

In order to confirm that the translational relationship of our findings in this murine model of HCC applies also to human tissues, we analyzed the expression of the CCK-BR using a human tissue microarray and human HCC cells in culture. The cholecystokinin-B receptor is not found in normal human liver tissue but becomes overexpressed in HCC [14]. Figure 6A shows representative images taken from normal human liver tissue and from human HCC tissues of different grades. The immunoreactivity of the CCK-BR is absent in the normal human liver but the intensity of the staining increases with the grade of tumor (Figure 6B; *p* = 0.0045).

#### 2.6.1. Upregulated Genes by Proglumide Using RNA Sequencing

A total of 138 genes had a log fold change >1.5 and were chosen for our extended analysis. Out of the 138 genes, 36 were cytoplasmic, all of which were enzymes, 16 belonged to the extracellular space, 27 were nuclear proteins, some of which were transcription regulators, and 33 were plasma membrane proteins that were either transporters or transmembrane receptors—mainly G-protein-coupled receptors and the remainder were classified as ‘Other.’ The top five statistically significant pathways and the upstream regulators are provided in Table 2.

The upstream regulators of these pathways include *DLGAP3, GFI1, ETV1, SPTAN1,* and *CXCL12*. The upregulation of these genes is involved in liver necrosis/cell death, HCC, and liver inflammation/hepatitis.

#### 2.6.2. Downregulated Genes by Proglumide in Human HCC Cells by RNA Sequencing

A total of 128 genes had a log fold change >1.5 and these were chosen for further analysis. Out of the 128 genes, 29 were cytoplasmic, with the majority being enzymes, 12 belonged to the extracellular space, 17 were nuclear proteins—some of which were transcription regulators, and 40 were plasma membrane proteins that were either ion channel proteins, transporters, or transmembrane receptors, mainly G-protein-coupled receptors and the remaining were classified as ‘Other’. Genes *GDF5, SLC18A2, SNCA, HSP70,* and *LOXL1* are upstream regulators of the downregulated genes. Of the significantly downregulated genes with proglumide in the human HepG2 cells, we found several consistent genes that were also identified with the mouse PCR array, including the following: extracellular matrix protein 2 (*ECM2*) mRNA was decreased 3.6-fold (*p* = 0.003) and collagen type 1α chain 1 *(COL1A1*) was decreased 2.6-fold (*p* = 1.87 × 10^−6^).

### 2.7. Differentially Expressed Genes in Human HCC Untreated or Treated with Proglumide by RNAseq

Total RNA was extracted from human HepG2 hepatocellular cancer cells treated with proglumide or control media in vitro and subjected to RNA sequencing to analyze selective genes and pathways altered by CCK-B receptor antagonism. Of the differentially expressed genes, 369 were upregulated (Appendix A) and 616 genes were downregulated (Appendix A) in human HCC cells treated with proglumide. These DEGs are shown in the volcano plot (Figure 7A) and a heat map (Figure 7B) portraying these differences.

## 3. Discussion

In this investigation, we described a novel target for treating HCC, the CCK-BR. The CCK-BR antagonist, proglumide, altered the HCC tumor microenvironment by decreasing fibrosis and increasing CD8+ T cells, rendering the tumor then responsive to immune checkpoint antibody therapy. The combination therapy with proglumide and the PD-1Ab exhibited an additive effect on mouse survival and, in part, this may be attributed to decreased expression of EMT-regulating genes. The CCK-BR protein is not found in the normal human liver but becomes expressed in liver cancer [20]. We previously showed in the Dt81Hepa1-6 murine HCC cell line that, even if both CCK-AR and CCK-BR are detected, ligand-stimulated growth is mediated by the CCK-BR [14]. There are two ligands for the CCK-BR: gastrin and CCK. Endogenous CCK blood levels are elevated with the consumption of a high-fat diet [21], especially saturated fats that are associated with fatty liver disease. Herein, we showed that RIL-175 cells express gastrin mRNA, suggesting perhaps that gastrin stimulates growth of HCC in an autocrine fashion. Caplin and colleagues [22] performed immunohistochemistry on HCC liver sections from 23 subjects for CCK-BR and pro-gastrin expression compared to 10 normal controls without HCC and found that 91% of the HCC tissues had CCK-BRs and 39% expressed pro-gastrin. In this current investigation, we showed that treatment of mice with a CCK-BR antagonist, proglumide, could inhibit RIL-175 tumor growth, implying that this drug is blocking the actions of endogenous CCK or gastrin at the receptor.

Although the RIL-175 cells express the PD-L1 receptor, treatment with PD-1Ab did not significantly alter tumor growth. The phenomenon of improving survival without visibly decreasing tumor size is a well-known occurrence with immune checkpoint antibody therapy and is due to the infiltration of T cells into the tumor [23]. The dose of the PD-1Ab we used at 50 µg weekly ×3 did not have any significant effect on tumor growth or the tumor microenvironment when administered alone. Shigeta et al. [24] also did not find a survival benefit or influx of CD8+ T cells in the RIL-175 HCC tumors, even when the PD-1Ab was administered at a higher dose and increased dosing schedule. In our work, a synergistic effect was noted when the PD-1Ab was administered in combination with proglumide in regard to the influx if CD8+ cells and in mRNA expression of differentially expressed genes regulating EMT. An additive effect was observed, with the combination therapy providing a survival benefit and decreased tumoral fibrosis, suggesting the importance of proglumide administration and its ability to improve the efficacy of PD-1Ab therapy.

Most liver cancers arise in the setting of fibrosis and cirrhosis and, thus, fibrosis is the major risk factor for development of HCC [25]. Furthermore, hepatic fibrosis impedes the penetration of T cells and therapies to the liver [26]. One remarkable finding in this study was that proglumide monotherapy decreased intratumoral fibrosis. CCK-BRs have been reported on fibroblasts [27] and in pancreatic stellate cells [28] but never in hepatic stellate cells. However, we have previously noted this anti-fibrotic effect of proglumide in mouse models of nonalcoholic steatohepatitis (NASH) [29] and also in human subjects with NASH treated with proglumide [30]. In this investigation, we further described the effects of proglumide on genes involved in fibrogenesis or fibrinolysis using RNA sequencing. Specific genes, such as matrix metallopeptidase 2 (*MMP2*), involved in the degradation of collagen 4 [31] were upregulated with proglumide therapy (Appendix A), whereas extracellular matrix protein 2 (*ECM2*) and (collagen type 1α chain (*COL1α1*) genes that promote fibrosis were significantly downregulated by proglumide (Appendix A). The influx of CD8+ T cells in tumors of mice treated with proglumide may be related to the decreased fibrosis observed.

Cancer invasion and metastasis are preceded by a phenotypic transformation in cancer cells through a process called epithelial-to-mesenchymal transition (EMT) [32]. Zeb1 or zinc finger E-box binding homeobox 1 is a transcription factor involved in EMT and functions as a repressor of the tumor suppressor protein E-cadherin [33]. Zeb2 is another zinc finger transcription factor that is increased in EMT and functions as a DNA-binding transcriptional repressor that interacts with activated SMADs [34]. Vimentin is a constituent of the intermediate filament family of proteins expressed in mesenchymal cells and a canonical marker of EMT [35]. Considerable amounts of data provide evidence suggesting EMT is a key process that is critical for immune resistance but also a potent driver for the activation of an immunosuppressive network within the tumor microenvironment (TME) [36]. Since EMT might contribute to immune escape, downregulation of genes implicated in EMT process (e.g., *Zeb1*, *Zeb2*, *Vimentin*, etc.) might be expected to improve the immuno-protective effect. Additionally, evidence also suggests that EMT might be playing an important role in fibrosis in liver cancer [37]; hence, there is a possibility that EMT and fibrosis may be important in influencing the response to combination of proglumide and the PD-1Ab.

The most important outcome from any cancer therapy is survival. In HCC-bearing mice, we found a significant survival benefit in mice treated with the combination of proglumide and the PD-1Ab. In addition to showing that the genes regulating EMT and fibrosis were decreased with this regimen, tumoral RNA analysis also demonstrated that selective genes involved in cell death, apoptosis, and ferroptosis [38] were significantly upregulated. A potential weakness of this investigation is that the effects of combination therapy were only studied in one in vivo model. However, the pathways and differentially expressed genes were consistent in two HCC models in vitro.

## 4. Materials and Methods

### 4.1. HCC Cell Lines

Murine RIL-175 hepatocellular carcinoma cells [39] were characterized and provided by colleague Dr. Tim Greten from the National Cancer Institute. The cells were grown in vitro in RPMI-1640 Medium supplemented with 10% fetal bovine serum and 1% penicillin/streptomycin in humidified air with 5% CO_2_. The RIL-175 cells were genetically authenticated and tested at the Animal Health Diagnostic laboratory, NCI Frederick, MD, USA, using the Molecular Testing of Biological Materials Mouse/Rat (MTBM-M/R) Test, and all the tests were negative for pathogens. Dt81Hepa1-6 murine HCC cells [16] were a gift from Dr. Marc Bilodeau (Montreal, QC, Canada) and are a highly metastatic cell line derived from the ATCC parent Hepa1-6 cells. The Dt81Hepa1-6 cells were tested and determined to be free of pathogens by the MU Research Animal Diagnostic Laboratory using the IMPACT I PCR Profile. Dt81Hepa1-6 cells were maintained in DMEM standard growth media supplemented with 10% fetal bovine serum and 1% pen/strep in humidified air with 5% CO_2._ The HepG2 human HCC cancer cell line was obtained from the ATCC through the Tissue and Cell culture repository of the Lombardi Comprehensive Cancer Center. This cell line has been used extensively to study human HCC and has been characterized as a hepatoblastoma-derived cell line [40]. These cells were maintained in DMEM media supplemented with 10% fetal bovine serum and 1% pen/strep in humidified air with 5% CO_2_.

### 4.2. Characterization of PD-L1 Receptor and Gastrin mRNA Expression in RIL-175 Cells

Normal mouse liver was harvested from 25 g C57BL/6 mice and RNA extracted to serve as a comparative normal control. RNA was extracted from RIL-175 cells using miRNeasy mini kit (Qiagen, Germantown, MD, USA). Synthesis of cDNA was performed using a qScript cDNA Synthesis Kit (Quanta Biosciences, Gaithersburg, MD, USA). Real-time PCR was performed using a Perfecta SYBR Green FastMix ROX kit (Quanta Biosciences) with an Applied Biosystems 7300 Real-Time PCR System machine to assess the expression of gastrin and PD-L1. Samples were subjected in triplicate to qRT-PCR, with an initial denaturation step of 95 °C at 3 min, followed by 40 cycles of 95 °C at 15 s and 60 °C at 1 min. The PCR primers used are shown in Table 3. Using in silico primer analysis, some cross-reactivity was noted between the *Cckbr* primer and murine *Zdhhc5* gene that codes for zinc finger DHHC-type palmitoyltransferase 5 that could potentially lead to off-target amplification. However, PCR dissociation curves revealed only one peak consistent with one amplicon (Appendix A).

### 4.3. Study Design for the Treatment of HCC Murine Tumors with Proglumide and PD-1Ab

All procedures performed involving animals were in accordance with the ethical standards of Georgetown University under the approved IACUC protocol number 2016-1193. Female C57BL/6 mice (*n* = 40) were injected with RIL-175 HCC cells (100,000) subcutaneously. After one week, when the mice had palpable tumors, they were divided into 4 groups with equal tumor size and *n* = 10 per treatment group: controls, proglumide (Tocris Bioscience, Minneapolis, MN, USA) at a concentration of 0.1 mg/mL in the drinking water; PD-1Ab (cat# BE0146) (Bio X cell, West Lebanon, NH, USA) (50 μg, ip, on days 7, 14, and 21), and combination therapy with a PD-1Ab and proglumide. Mice were euthanized when tumors reached a diameter of 20 mm according to IACUC guidelines.

### 4.4. Analysis of the HCC Tumor Microenvironment for CD8+ T Cells and Fibrosis

Tumors were excised and fixed in 4% paraformaldehyde and paraffin embedded. Tissues sections of 5 µm were prepared and mounted on slides. Tumors were stained with a CD8 antibody (cat # 98941) (Cell Signaling, Danvers, MA, USA). The slides were scanned using an Aperio GT450 machine and images were captured with software from Aperio Image Scope version 12.4.6 and the number of immunoreactive CD8+ T cells was analyzed using Image J computer software (https://imagej.nih.gov/ij/download.html). Tissue sections were also stained with Masson’s trichrome for analysis of fibrosis in the tumor microenvironment.

### 4.5. Examination of Differentially Expressed Genes (DEGs) in Murine HCC Tumors and Cells

RIL-175 tumor tissues were homogenized in QIAzol Lysis reagent (Qiagen) solution, followed by RNA extraction using miRNeasy mini kit (Qiagen). cDNA was prepared from purified RNA using RT2 First Strand Kit and N = 4 samples were combined from the control group and *n* = 4 samples were combined from tumors of the combination-treated groups. The combined cDNA from the control and combination-treated groups was subjected to Qiagen RT2 Profiler PCR Array Mouse Liver Cancer (Cat. #330231) (Qiagen) using an Applied Biosystems 7300 Real-Time PCR System machine to assess the expression of 84 genes. In order to further evaluate and validate genes identified with the PCR array, RNA was extracted from each of the control and treatment groups (N = 4 tumors/group) and subjected to qRT-PCR to specifically evaluate genes from the epithelial-to-mesenchymal (EMT) pathway, including *Zeb1, Zeb2, Vimentin*, and two genes involved with fibrosis: *Col1α1 and Acta2*.

Real-time PCR analysis was carried out, with an initial denaturation step at 95 °C 10 min, followed by 40 cycles of 95 °C at 15 sec and 60 °C at 1 min. Data were collected by the PCR software at the end of each cycle. HPRT served as the normalizer control. The primers are indicated in Table 4. A dissociation curve analysis of PCR products was carried out to confirm the specificity of amplification (Appendix A). The relative differences between two groups were calculated using ∆∆CT method. The gene expression data using quantitative RT-PCR (qRT-PCR) are relative expression data and were calculated with ΔΔCt method using raw Ct data. The raw Ct data and detailed calculation of relative expression data have been provided in a table as a part of Appendix A. The relative expression for a gene in the treated group was compared with the control group. The first sample in the control group was arbitrarily assigned a value of 1, so all other data samples (control as well as treated) were compared to it accordingly. This provides appropriate relative data across the data points. Significance between data from two groups was determined with Prism GraphPad Version 9.0 (Graph Pad Software, San Diego, CA, USA) by using unpaired Student *t*-test (*p* ≤ 0.05). *p* ≤ 0.05 is considered statistically significant.

Murine Dt81Hepa1-6 cells were grown in vitro to log phase, then were treated with media alone (control) or proglumide (20 nM and 40 nM) for either 8 or 24 h. RNA was extracted from cells as above and subjected to qRT-PCR with the murine primers in Table 4 and analyzed using Prism GraphPad 9.0 (Graph Pad Software) as above. Using in silico primer analysis, some complementary results were identified between the nucleotides for *Col1α1* and *Foxr2* (Mus musculus forkhead box R2) gene. However, the PCR dissociation curves revealed only one peak (Appendix A) and *Foxr2* gene is not expressed in mouse liver.

### 4.6. Immunohistochemistry for the CCK-BR in Human Liver Cancer

CCK-BR protein expression in human liver tissues and human HCC was evaluated using a tissue microarray (Cat# BC03117) (US Biomax, Rockville, MD, USA). This array contained 49 cases of HCC and 8 tissues from normal human liver without cancer. After deparaffination and antigen retrieval procedures, the array was reacted with a goat polyclonal CCK-BR primary antibody, ab7707 (Abcam, Eugene, OR, USA) at 1:200 overnight at 4 °C. The intensity of CCK-BR immunoreactivity was scored (0–5) by our team liver pathologist from no antibody staining (0) to intense staining (5).

### 4.7. Differentially Expressed Genes by RNA Sequencing in Human HCC Cells Exposed to Proglumide

Human HepG2 cells were obtained from the American Type Collection (ATCC, Manassas, VA, USA) and grown in vitro in DMEM in 6-well plates to log-phase growth. HepG2 cells were treated with proglumide (20 nM) or media alone (Control) for 24 h. We have previously found that a dose between 20 and 40 nM is the most effective at blocking the CCK-BR [14,41] and proglumide has a T1/2 of 24 h. RNA was extracted as above, and the RNA integrity and quantitation were assessed using the RNA Nano 6000 Assay Kit of the Bioanalyzer 2100 system (Agilent Technologies, Santa Clara, CA, USA) and RIN values > 9.0 were used. Library preparation for transcriptome sequencing was performed by Novogene Co., Ltd. (Sacramento, CA, USA) with 1 µg RNA per sample, as previously described by our group [19].

### 4.8. Statistical Analysis

Statistical analyses were conducted with GraphPad Prism v.9, Minitab V.19.0 (Graph Pad Software), and the R statistical programing language. We have determined that at least *n* = 8–10 mice in each treatment group were required to reach a power of 0.80 from our prior studies between treatment and control [19,42]. Real-time PCR results were analyzed using a Student’s T-test on the normalized mean ΔCT (the difference between the cycle counts of the gene of interest minus the count of an endogenous control) values for each group, with Bonferroni corrections applied to adjust for multiple comparisons.

The human RNA sequencing data were analyzed with Qiagen Ingenuity Pathway Analysis (IPA) for bioinformatics analysis, which included canonical pathway analysis, disease and function, regulator effects, upstream regulators, and molecular networks. IPA uses a network generation algorithm to segment the network map between molecules into multiple networks and assign scores for each network. The network overlap function was used to generate a network that connects pathways that had overlapping genes.

## 5. Conclusions

The CCK-BR antagonist, proglumide, significantly alters murine HCC tumoral fibrosis by downregulating fibrosis-associated genes in the tumor microenvironment. In addition, proglumide monotherapy alters HCC intratumoral immune cell signature. When proglumide is administered together with PD-1Ab, there appears to be a synergistic effect on tumoral immune cell signature and differentially expressed genes, resulting in a survival benefit. Future studies using proglumide in human subjects with HCC that are eligible for immune checkpoint antibody therapy may be advantageous.

## 6. Patents

Georgetown University holds a patent titled ‘Treating Cancer with a CCK Receptor Inhibitor and an Immune Checkpoint Inhibitor.’ US Patent Application #16/493,882 filed on 13 September 2019 and issued as a US patent No. 11,278,551 on 22 March 2022.

## Figures and Tables

**Figure 1 ijms-24-03625-f001:**
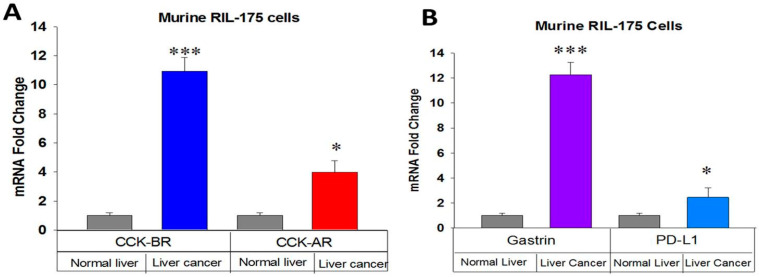
Characterization of murine RIL-175 HCC cells in vitro. (**A**) Cholecystokinin-B receptor (CCK-A) and cholecystokinin-A receptor (CCK-B receptor) mRNA expression is significantly increased in RIL-175 HCC cells by qRT-PCR compared to normal mouse liver. (**B**) Gastrin and PD-L1 expression by qRT-PCR are increased in RIL-175 mouse HCC cells compared to normal mouse liver. *** *p* < 0.001; * *p* < 0.05.

**Figure 2 ijms-24-03625-f002:**
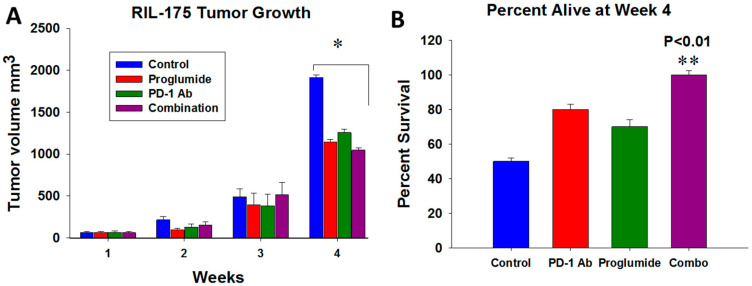
Effects of proglumide and PD-1Ab alone or in combination (Combo) on RIL-175 HCC tumor growth in vivo. (**A**) Tumor volumes after subcutaneous injection of 100,000 RIL-175 cells in C57BL/6 mice show a significant change in tumor volume when both proglumide and PD-1Ab were administered compared to control mice (* *p* < 0.05). (**B**) Percentage of mice alive after 4 weeks of treatment show a significant survival benefit in the mice treated with the combination therapy (** *p* < 0.01).

**Figure 3 ijms-24-03625-f003:**
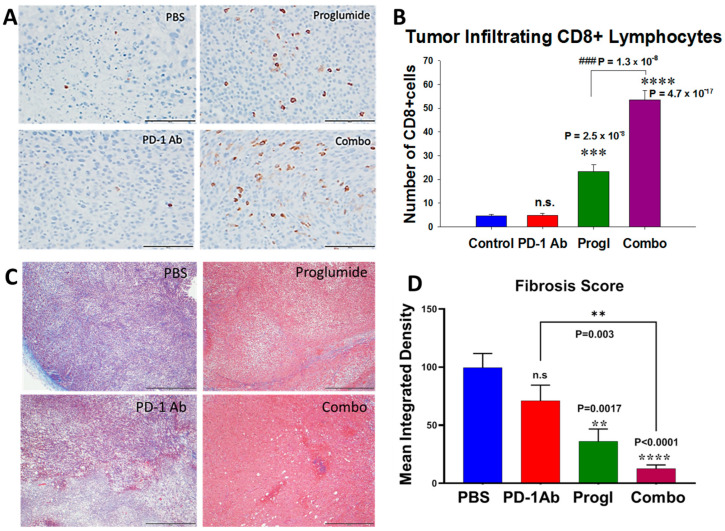
Effects of proglumide and the PD-1Ab on the HCC tumor microenvironment. (**A**) RIL-175 tumors were sectioned and reacted with an antibody to stain CD8+ T cells (Cell Signaling; Cat # 98941). A representative image of the immunohistochemistry from each group is shown. (**B**) Mean values ± SEM for number of CD8+ tumor infiltrating lymphocytes is shown. Only proglumide and the combination (Combo)-treated group had more tumoral CD8+ T cells than controls. The combination therapy was synergistic. (* compared to PBS and # compared to proglumide monotherapy vs. combination therapy.) Bar = 100 µm. (**C**) Masson trichrome staining of fibrosis in RIL-175 tumor sections is shown with representative images from each treatment group. (**D**) Tumoral fibrosis by Masson’s trichrome stain was analyzed by computerized densitometry. Only tumors of proglumide-treated mice had less fibrosis compared to controls. (*p* values as shown.) Bar = 500 µm; n.s. = not significant.

**Figure 4 ijms-24-03625-f004:**
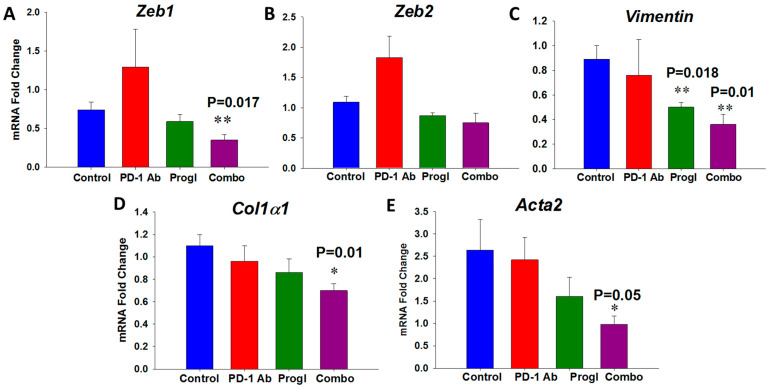
Evaluation of selective genes from all four groups of RIL-175 tumors involved in epithelial-to-mesenchymal transition (EMT) and fibrosis. (**A**–**C**) mRNA fold change by qRT-PCR of genes involved in EMT shows that mice treated with the combination (Combo) had significantly decreased expression of (**A**) *Zeb1*; (**B**) *Zeb2*; and (**C**) *Vimentin.* (**D**) Expression of *Col1α1* that codes for collagen-α1 is significantly decreased in tumors of mice treated with the Combo therapy. (**E**) Expression of *Acta2*, the gene that codes for α-smooth muscle actin from activated fibroblasts, was also significantly decreased in tumors of mice treated with Combo.

**Figure 5 ijms-24-03625-f005:**
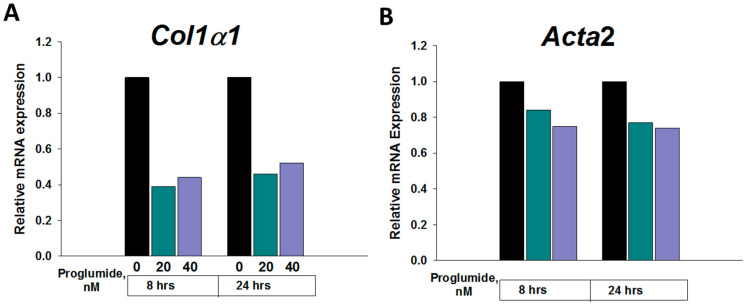
Relative mRNA expression of the fibrosis genes in Dt81Hepa1-6 HCC. Cells were untreated (0) or treated with proglumide (20 or 40 nM) for either 8 or 24 h. (**A**) Relative mRNA expression by qRT-PCR of *Col1α1* and (**B**) Relative mRNA expression by qRT-PCR of *Acta2*.

**Figure 6 ijms-24-03625-f006:**
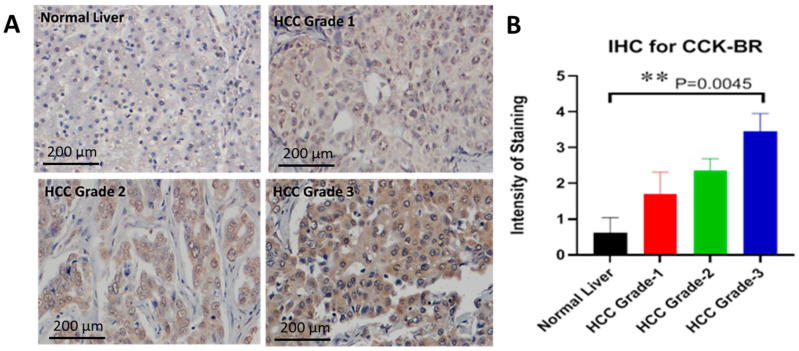
CCK-BR immunoreactivity in a human liver and HCC tissue array. (**A**) Representative photos from a human liver tissue microarray of normal liver and HCC tissues stained for immunoreactivity for the CCK-BR. (**B**) Scoring of the human microarray show the absence of staining in normal human liver tissue and positive CCK-BR staining increases with the tumor grade.

**Figure 7 ijms-24-03625-f007:**
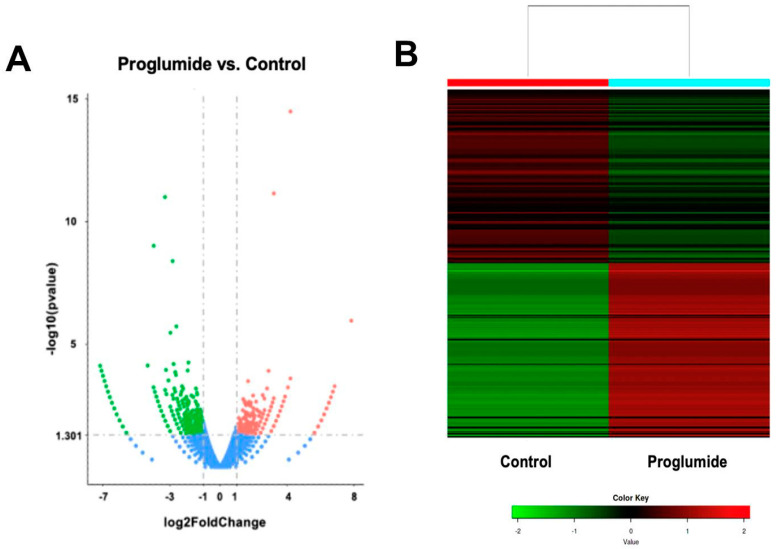
Differentially expressed genes in proglumide-treated human HepG2 HCC cells. (**A**) Volcano plot shows 369 upregulated (pink) and 616 downregulated genes (green) with proglumide as determined by RNA sequencing. Genes not significantly changed are blue. (**B**) Heat map of up- and downregulated genes.

**Table 1 ijms-24-03625-t001:** Description of the top 10 DEGs from tumors of control versus Combo-treated mice and the PubMed reference number.

Gene	Protein	Function	Expression in Combination Therapy	PubMed
*Igfbp3*	insulin-like growth factor binding protein 3	Metastases suppression gene	Up	21697285
*Ptgs2*	prostaglandin-endo-peroxide synthase 2	Downstream marker of ferroptosis, cell death	Up	30925886
*Tert*	Telomerase reverse transcriptase	Promotes immortalization by protecting telomeres	Up	30447097
*Col1αα1*	Collagen 1-alpha	Makes collagen/fibrosis/promotes metastases	Down	31181620
*Zeb2*	Zinc Finger E-Box Binding Homeobox 2	Regulator of EMT	Down	30157906
*Zeb1*	Zinc Finger E-Box Binding Homeobox 1	Regulator of EMT	Down	25607528
*Vimentin*	fibroblast intermediate filament,	Promotes EMT	Down	25965826
*Lef1*	Lymphoid enhancer binding factor 1	Promotes EMT & WNT signaling	Down	28670499
*Acta2*	Alpha-smooth muscle actin	EMT and fibrosis	Down	23128109
*Sfrp2*	Secreted frizzled-related protein 2	Promotes EMT	Down	28218291

**Table 2 ijms-24-03625-t002:** Top statistically significant pathways computed using IPA software tool for upregulated genes.

Pathway	*p*-Value
Phagosome Formation	5.88 × 10^−3^
Pyroptosis Signaling Pathway	1.55 × 10^−2^
Macrophage migration inhibitory factor (MIF)-mediated Glucocorticoid Regulation	1.74 × 10^−2^
G-Protein-Coupled Receptor Signaling	1.78 × 10^−2^
Inhibition of Matrix Metalloproteases	2.02 × 10^−2^

**Table 3 ijms-24-03625-t003:** PCR primers.

mGene	5′-Forward Primer-3′	5′-Reverse Primer-3′
*Cckar*	CTTTTCTGCCTGGATCAACCT	ACCGTGATAACCAGCGTGTTC
*Cckbr*	GATGGCTGCTACGTGCAACT	CGCACCACCCGCTTCTTAG
*Cd274*	TGCGGACTACAAGCGAATCACG	CTCAGCTTCTGGATAACCCTCG
*Gast*	CCCAGGGTCCTCAACACTTC	GCCAAAGTCCATCCATCCGT
*Hprt*	TCCTCCTCAGACCGCTTT	TTTTCCAAATCCTCGGCATAATG

*Cckar* = cholecystokinin A receptor; *Cckbr* = cholecystokinin B receptor; *Cd274*: mouse gene that codes for PD-L1 programed death ligand-1; *Gast* = gastrin; and *Hprt* = hypoxanthine–guanine phosphoribosyltransferase (normalizer gene). *Gast* primers were designed and validated by GeneCopoeia, Inc. (Rockville, MD, USA) (Cat #MQP029209 for target NM_010257.4).

**Table 4 ijms-24-03625-t004:** PCR primers for DEGs.

mGene	5′-Forward Primer-3′	5′-Reverse Primer-3′
*Zeb1*	GCTGGCAAGACAACGTGAAAG	GCCTCAGGATAAATGACGGC
*Zeb2*	GGGACAGATCAGCACCAAAT	GACCCAGAATGAGACAAGCG
*Vimentin*	TCCACACGCACCTACAGTCT	CCGAGGACCGGGTCACATA
*Col1α1*	CGCCATCAAGGTCTACTG	ACGGGAATCCATCGGTC
*Acta2*	TGCCGAGCGTGAGATTGT	CCCGTCAGGCAGTTCGTAG

## Data Availability

Data are available in the Appendix A and upon request from the primary author.

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
