# Peer review of "Treatment with a Cholecystokinin Receptor Antagonist, Proglumide, Improves Efficacy of Immune Checkpoint Antibodies in Hepatocellular Carcinoma"

_ijms, 2023, doi:10.3390/ijms24043625_

Round 1

Reviewer 1 Report

In Shivapurkar et al., the authors describe the discovery of CCK-BR as a potential drug target for Hepatocellular carcinoma (HCC). The authors demonstrated higher gene expression of CCK-BR in liver cancer cells than in the normal liver. The authors have also demonstrated that Gastrin is upregulated in liver cancer cells. Furthermore, the authors showed that treatment with Proglumide, an antagonist of CCK-BR, in combination with PD-1 results in lower tumor growth, more t-cell infiltration, lower fibrosis, and lower expression of oncogenes. Moreover, the authors performed RNA-Seq on Human HepG2 cell lines treated with Proglumide vs. untreated and showed upregulation of pathways associated with immune responses. Overall, the study is promising. However, in its current format, we cannot assess the validity of the results, and the investigation does not provide mechanistic details on how proglumide treatment can have a therapeutic effect on HCC. I have the following concerns and recommendations:

1- Rephrase the statement in line 21, "but the response remains only 25-40%".

2- According to in-silico PCR, Cck-br primer is not specific. Replace primer since it is an essential point.

3- PD-L1 and Gastrin primers used in the study do not amplify the indicated genes as per in-silico PCR. The qPCR results for these genes can not be trusted.

4- Col1a1 primer is not specific. Replace the primer or remove the result.

5- Please use official gene symbols; for the mouse experiments, use proper capitalization of letters.

6- In figure 4A, the standard deviation is absent. 

7- Lines 190 - 197 are not visible, making it hard to understand and assess the RNA-Seq results. 

8- Instead of just performing DGE for RNA-Seq, the authors should predict significant complexes downregulated upon proglumide treatment using analyses such as STRING, Cytoscape, and MCODE.

9- To elevate the study's significance, I recommend using the published Single-Cell Atlas to understand which cell types in the tumor microenvironment express Cck-br.

Reviewer 2 Report

In this manuscript, the authors present data suggesting that proglumide, a cholecystokinin receptor agonist, could be proposed for treating patient with HCC in combination with checkpoint inhibitors. To support this proposal, they demonstrate increase of immune-infiltration in tumors and decrease of fibrosis in TME after treatment with proglumide.

Pathology, treatments, and the rational for using a CCK-BR antagonist are well presented in Introduction. Experiments are relatively well explained and data are clear. However, some modifications/additions are need for improving the manuscript.

1/ “anti-PD1 antibody” and not « PD1 antibody”

2/ It is not possible to use the term “synergy” here. That refer to well defined phenomenon and need a specific study like the Combination Index calculation for really determine if there is a synergy. Another term could be “”additive, “cumulative or “positively combined” effect.

3/ It is not surprising to see an effect on survival and not on tumor growth here. This phenomenon was well described with checkpoint inhibitors, even in clinic. The authors could go back and refer to first studies on melanoma when an increase of tumor volume was observed before a drastic reduction. Initially, tumor volume does not change or even increase because of lymphocytes infiltration. The authors should mention this fact.

4/ In vivo experiment is performed with one model only. The author should notice this point in the text to mentioned that results have to be used with cautious because of that. Hopefully, the other experiments with human material confirm in vivo data and reinforce the message of the manuscript.

5/ the fact to use only one cell line for transcriptomic study is more problematic. This study should be reinforced at least with another human HCC cell line in order to discuss only variation with a consensus between the 2 cell lines. It is possible here that some variations are specific to the HepG2 cell line, they complicate the interpretation and produce some errors.

The establishment of the protocol is not completely clear as well: why proglumide at 20nM ?It is certainly possible to see different transcriptomic variations by using different concentrations. Similar question for the time: why a determination at 24h only ? Early variation are usually observed at 6, 8 or 12h and observation at 24h could reflect variation of secondary genes in addition to modulation of primary gene.

The choice of conditions has to be better explained and this experiment really needs to be reinforced by another cell line and other conditions to be convincing.

Minor point: table in Figure 4B is really difficult to read and should be bigger.

Author Response

Reviewer 2

Comments and Suggestions for Authors

In this manuscript, the authors’ present data suggesting that proglumide, a cholecystokinin receptor agonist, could be proposed for treating patient with HCC in combination with checkpoint inhibitors. To support this proposal, they demonstrate increase of immune-infiltration in tumors and decrease of fibrosis in TME after treatment with proglumide.

Pathology, treatments, and the rational for using a CCK-BR antagonist are well presented in Introduction. Experiments are relatively well explained and data are clear. However, some modifications/additions are need for improving the manuscript.

Comment 1: “anti-PD1 antibody” and not « PD1 antibody”

Response 1: This nomenclature was changed throughout at the reviewer’s request but either wording is appropriate.

Comment 2: It is not possible to use the term “synergy” here. That refers to well defined phenomenon and need a specific study like the Combination Index calculation for really determine if there is a synergy. Another term could be “”additive, “cumulative or “positively combined” effect.

Response 2: The definition of “synergy or synergistic effect” is relating to the interaction or cooperation of two or more substances, or other agents to produce a combined effect greater than the sum of their separate effects.

Below is a point by point calculation to show that ALL the values showed synergy with the combination therapy except survival and fibrosis. (These two were additive). The text has been revised to clarify these differences.

  • With the CD8 cell numbers in tissue: proglumide monotherapy (23.35) + anti-PD1 Ab monotherapy (4.92) = 28.27- however when the two compounds were combine the result was greater than the sum = (53.56). This is synergy.
  • Fibrosis score was decreased by monotherapy anti-PD1 (28.28) + Proglumide (63.33) = 91.61, but the combination effect was decreased by 86.8. This is additive.
  • Survival % increase over control: anti-PD1 (30%) + Proglumide (20%) = 50% but the combination is also 50% more so this is additive.
  • aSMA expression is decreased anti-PD1 (0.22) + proglumide (1.04)= 1.26; but when combined the decrease was 1.62 fold. This is synergy.
  • COL1a expression was decreased anti-PD1 (.14 fold) + proglumide (.24) fold =0.38 but the combination decreased expression 0.4-fold. This is synergy.
  • Vimentin decreased expression anti-PD-1 (0.13) + proglumide (.39)= 0.52 but the combination decreased expression 0.53. This is synergy.
  • Zeb1 expression actually increased with anti-PD1 so from control (+0.55)+ proglumide (-0.15)= +0.4, but the combination was decreased -0.35. This is synergy.
  • Zeb2 expression actually increased with anti-PD1 so from control (+0.74) + proglumide (-0.22)= +0.52, but the combination therapy was -0.34. This is synergy.

Comment 3. It is not surprising to see an effect on survival and not on tumor growth here. This phenomenon was well described with checkpoint inhibitors, even in clinic. The authors could go back and refer to first studies on melanoma when an increase of tumor volume was observed before a drastic reduction. Initially, tumor volume does not change or even increase because of lymphocytes infiltration. The authors should mention this fact.

Response 3: We are well aware of this phenomenon and have included reference 17 explaining that the lack of radiographic or visible tumor volume with a survival benefit is well known with this treatment.

Comment 4. In vivo experiment is performed with one model only. The author should notice this point in the text to mention that results have to be used with cautious because of that. Hopefully, the other experiments with human material confirm in vivo data and reinforce the message of the manuscript.

Response 4: We have added this to the end of the discussion regarding the potential weaknesses of this work.

Comment 5: the fact to use only one cell line for transcriptomic study is more problematic. This study should be reinforced at least with another human HCC cell line in order to discuss only variation with a consensus between the 2 cell lines. It is possible here that some variations are specific to the HepG2 cell line, they complicate the interpretation and produce some errors.

Response 5: In the revised manuscript we have repeated qRT-PCR to evaluate transcriptomic data in a second HCC cell line Dt81Hepa1-6 n vitro. We believe any differences may be related to the RIL-175 analysis was from tumors that included the immune cells and tumor microenvironment genes, whereas the in vitro data was just from the HCC cells. Using the 2nd cell line we now have confirmatory gene analysis data compatible with the other in vitro cell line and with many of the genes from the HCC tumor.

Comment 6: The establishment of the protocol is not completely clear as well: why proglumide at 20nM ? It is certainly possible to see different transcriptomic variations by using different concentrations. Similar question for the time: why a determination at 24h only ? Early variation are usually observed at 6, 8 or 12h and observation at 24h could reflect variation of secondary genes in addition to modulation of primary gene. The choice of conditions has to be better explained and this experiment really needs to be reinforced by another cell line and other conditions to be convincing.

Response 6: We have previously tested various doses of proglumide in vitro, in vivo and in humans. And we know that 2 nM is not effective, 20-40nM is effective at blocking the receptor (DOI: 10.1158/1940-6207.CAPR-20-0220; DOI: 10.1016/s0016-5085(88)80075-1 and 200nM is toxic (manuscript in draft). In mice we use it in drinking water at 0.1mg/ml. We have even done a dose finding response in humans subjects (doi: 10.1002/cpt.2745). But in response to your comment, we have repeated the transcriptomic data in another in vitro cell line (Dt81Hepa1-6) and we have added the significant results to the revised manuscript in Figure 6 and in Supplemental materials Table T2. In this data in the revision, we show the effects of proglumide at 20 and 40 nM and also at 8 and 24 hrs.

Comment 7: Minor point: table in Figure 4B is really difficult to read and should be bigger.

Response 7: the figure changed to a Table (Table 1) and is much more legible.

Reviewer 3 Report

Very interesting paper, however, a few details need to be clarified:

1. In the description of the results in section 2.1 and in Figure 1, the presented results refer to "normal liver" and "liver cancer". There is no mention of "normal liver" in the methodology in section 4.1.

2. Please specify what cells/tissues are being compared, whether the cell lines have authentication, or in the case of mouse liver cells, the bioethics commission has approved the study. How much RNA were taken to the reverse transcription?  In addition, please prepare descriptions of figures and their legends in such a way that it is clear what is presented.

3. Please also provide access to the raw results from the expression analysis using the qRT-PCR method and explain the method abbreviation. As I understand it, "q" stands for quantitative. Please provide information about the standard curve and raw results from qRT-PCR method.

4. In the description of the qRT-PCR method (line 295) there is a statement "in triplicate or 40 cycles at 60oC". This is probably a mistake. Please correct. How was the specificity of the reaction with SYBR Green tested?

5. The description of all methods needs to be supplemented so that it is clear how the experimental part was carried out.

Author Response

Reviewer 3

Very interesting paper, however, a few details need to be clarified:

  1. In the description of the results in section 2.1 and in Figure 1, the presented results refer to "normal liver" and "liver cancer". There is no mention of "normal liver" in the methodology in section 4.1.

Response: The normal liver was added to the methods

  1. Please specify what cells/tissues are being compared, whether the cell lines have authentication, or in the case of mouse liver cells, the bioethics commission has approved the study. How much RNA were taken to the reverse transcription?  In addition, please prepare descriptions of figures and their legends in such a way that it is clear what is presented.

Response: In the revised Materials and methods section we have added an entire section on description of the 3 cell lines used in this investigation. The cell line RIL-175 was authenticated and pathogen tested and IACUC approved. This was in the methods section. We extracted total RNA from biospecimens. Stock RNA working solutions were prepared (200ng/ul), and 1600ng (1.6ug) was used to prepare c-DNA. In response to reviewer’s suggestion the description of figures and their legends were prepared in a way that it is clear what is presented.

  1. Please also provide access to the raw results from the expression analysis using the qRT-PCR method and explain the method abbreviation. As I understand it, "q" stands for quantitative. Please provide information about the standard curve and raw results from qRT-PCR method.

Response: The gene relative expression data were calculated using DDCt method using raw Ct data. Thus it does not involve a standard curve. The raw Ct data and detail relative expression data have now been provided in a Table as a part of supplemental data. The relative expression for a gene in treated group was compared with control group. The first sample in treated group was arbitrarily assigned a value of 1 so all other data samples (control as well as treated) are compared to it accordingly thus provide appropriate relative data across the data points. Thus these data show how we calculated relative expression from raw Ct values for individual data points.  Since we used the SYBRGreen methodology we have included dissociation curves in the supplementary data section which show a single peak indicative of single amplicon (specificity of amplification) in case of all the genes analyzed.

  1. In the description of the qRT-PCR method (line 295) there is a statement "in triplicate or 40 cycles at 60oC". This is probably a mistake. Please correct. How was the specificity of the reaction with SYBR Green tested?

Response; This was a typo and corrected. Melt curves are provided and raw data in the revised supplemental data.

  1. The description of all methods needs to be supplemented so that it is clear how the experimental part was carried out.

Response: In the revised Materials and methods, we have provided an expanded description of methodology as requested.

Round 2

Reviewer 1 Report

Please see my responses in the attached file.

Reviewer 3 Report

The authors did not include raw data from the real time PCR method. Only sequencing data are included in the supplementary materials. The work contains inaccuracies and a lot of imprecise information regarding the methodology. There are also no statistical analysis from real-time PCR experiments. There is also no information about the approval of the bioethics committee for experiments on animals.

Author Response

The PCR methods are now provided in the revision manuscript Sections 4.2 and 4.5 and are highlighted in the attachment for easy identification.

Round 3

Reviewer 1 Report

In this round of revision, the authors have demonstrated that the qPCR amplicons are specific in their experiments.

Just as a correction to the response letter, the Zdhhc5 gene has only five mismatches and not five matches. Which means it is a potential off-target. So that's why it gets pointed out by NCBI-primer blast. However, the amplification plot is convincing. But I strongly recommend adding a point regarding this in the method section in case other readers use this primer sequence for a cell line that expresses the Zdhhc5 gene.

Similarly, the Foxr2 gene has only four mismatches, not four matches. My recommendation is the same as the point mentioned above.

My other concerns have been addressed. I recommend the publication of the manuscript after adding the note about Zdhhc5 and Foxr2 in the methods section.

I want to congratulate the authors for the interesting study.

Author Response

Reviewer 1

Just as a correction to the response letter, the Zdhhc5 gene has only five mismatches and not five matches. Which means it is a potential off-target. So that's why it gets pointed out by NCBI-primer blast. However, the amplification plot is convincing. But I strongly recommend adding a point regarding this in the method section in case other readers use this primer sequence for a cell line that expresses the Zdhhc5 gene.

Response:  We have added this gene in the methods by the Cckbr primer explaining potential cross-reactivity.

Similarly, the Foxr2 gene has only four mismatches, not four matches. My recommendation is the same as the point mentioned above.

Response:  We have added this gene in the methods by the Col1a1 primer explaining potential cross-reactivity.

My other concerns have been addressed. I recommend the publication of the manuscript after adding the note about Zdhhc5 and Foxr2 in the methods section.

I want to congratulate the authors for the interesting study.

Response: Thank you

Reviewer 3 Report

Unfortunately, the data in Supplementary Table 1 is not clearly presented, in particular how the fold change was calculated. The information in this table is not covered in any of the results paragraphs, no reference to this table in the manuscript, inconsistently used abbreviations for p-value, and incorrectly used nomenclature for writing gene names. 

Author Response

Unfortunately, the data in Supplementary Table 1 is not clearly presented, in particular how the fold change was calculated. The information in this table is not covered in any of the results paragraphs, no reference to this table in the manuscript, inconsistently used abbreviations for p-value, and incorrectly used nomenclature for writing gene names. 

Response: Reference to Supplementary Table 1 in the text was accidentally left out. We now refer to this table in the Results section 2.4 and Methods Section 4.5. The data in the Supplementary Table1 is now clearly presented. How the fold change was calculated has been described in detail in Materials and Methods Section 4.5 and also briefly referred to in the supplemental Table1. The abbreviations and gene nomenclatures are appropriately corrected.